# Match analysis and probability of winning a point in elite men's singles tennis

Iván Prieto-Lage[1]*, Adrián Paramés-González[1], Daniel Torres-Santos[1], Juan Carlos Argibay-González[1], Xoana Reguera-López-de-la-Osa[2], Alfonso Gutiérrez-Santiago[1]

**1** Faculty of Education and Sport, Observational Research Group, University of Vigo (Spain), Vigo, Spain, **2** Galicia Sur Health Research Institute (IIS Galicia Sur), Education, Physical Activity and Health Research Group (Gies10-DE3), SERGAS-UVIGO, Vigo, Spain

* ivanprieto@uvigo.es

**Data Availability Statement:** All relevant data are within the paper and its Supporting Information files.

## Abstract

Notational analysis and new technologies have allowed a better understanding of tactical actions in tennis. In particular, the combined analysis of different variables affecting performance is necessary to understand the relationships between actions in competition. The aim of this research was to analyse the probability of winning a point in men's professional tennis based on the most relevant variables affecting performance in this sport. A total of 4,669 points were analysed on three different court surfaces from the final rounds (from the quarter-finals onwards) of three of the four Grand Slam tournaments in the 2021 season. An observational methodology was applied. Different analysis techniques were used to obtain the results: descriptive and chi-square with a significance level of $p<0.05$. First serve effectiveness (point won) was 69% on clay, 75% on grass and 75% on hard court. Second serve effectiveness (point won) was around 55% regardless of the surface. The majority of points, between 65% and 77% depending on the court surface, ended with a short rally (between one and four shots). Approximately 80% of the points played with first serve and short rally were won by the serving player. With first serve and medium length rallies, the probability of winning the point is similar between server (range 49–55%) and receiver on any court surface. The study reveals a set of patterns (based on the combination of information from the variables analysed) that determine the probability of winning a point. Descriptive data from this research could help coaches and players on match strategy at the highest levels of elite men's single tennis.

## Introduction

Match analysis in tennis is a topic of interest to sport performance researchers, as can be seen in the numerous investigations that have proliferated recently [1–4]. Like many sports, tennis has a long history of qualitative and quantitative performance analysis [5], most of it predating the automated data collection systems recently seen in modern sport. The automation of data collection and reporting systems has recently accelerated due to new analytical tools and a growing awareness in sport/coaching of the value of analytics [1,5].

**Funding:** This study was funded by the Ministerio de Cultura y Deporte (https://www.culturaydeporte. gob.es/portada.html), Consejo Superior de Deportes (https://www.csd.gob.es/es) and European Union (https://european-union.europa. eu/index_es) under Project "Integración entre datos observacionales y datos provenientes de sensores externos: Evolución del software LINCE PLUS y desarrollo de la aplicación móvil para la optimización del deporte y la actividad física beneficiosa para la salud (2023)" EXP_74847 to IPL and AGS. The funders had no role in study design, data collection and analysis, decision to publish, or preparation of the manuscript.

**Competing interests:** The authors have declared that no competing interests exist.

The introduction of new technologies in this sport is generating a large amount of data when analysing the competition. It is now possible to access statistical information of different types without difficulty. The Association of Tennis Professionals (ATP), the Women's Tennis Association (WTA) and the International Tennis Federation (ITF) provide statistics on the most relevant competitions through their partnerships with Infosys, SAP and Slam Tracker [6]. This data can help tennis players' teams focus on the search for meaningful variables for performance enhancement. It can also be useful to prepare match strategies and help decision making during the match [5,7]. New analytical techniques have improved the interpretation of sports data and provided more meaningful insights [8]. This data analysis provides insight into the variables that have the greatest influence on the final result of the match [9].

The analysis of the match has allowed the extraction of information that affects the performance related to the type of game [10,11], physical condition [12], technical [13], tactical [2] and psychological [14] aspects. Most researchers agree that the serve is the most decisive technical action of the match [3,15], due to the fact that the server has the first opportunity to win the point through an ace, an error by the opponent or to obtain a tactical superiority that puts them in an advantageous situation during the first shots [16], where it has been shown that most of the points are finished [17]. The first serve (putting the ball into play) barely varies according to the surface (between 62–64% depending on the court surface), although differences have been found in terms of achieving a direct serve (ace) [3]. Numerous studies have shown a clear difference in the number of points won depending on whether the first or second serve is made. While it is true that the player wins 69–75% of the points on the first serve, the percentage drops to 47–57% on the second serve, depending on the surface [3,13,18,19].

Another key factor is the type of surface on which the match is played [20]. The ITF classification divides court surfaces into 5 categories (slow, medium slow, medium, medium fast and fast). In the case of slow surfaces (e.g.: clay), the ball will present a higher coefficient of friction, a decreased horizontal speed and an increased bounce height; in the case of fast surfaces (e.g.: grass or hard track) the opposite will be the case [21]. In each tournament the court surface is different [22] and even when matches are played on natural surfaces (grass and clay), it must be taken into account that the surface varies as the days of competition progress, which can cause points to be played faster or slower [23] (without considering weather factors as well). According to the experts, the type of surface affects whether the point is easier for the server to win [20]. Another aspect that clearly influences performance in the sport is the rally length. Researchers have corroborated that most rallies end before five shots, which is more accentuated on fast surfaces [24,25]. This aspect of the game has a significant impact on match strategy [10,26]. Another determining factor is the moment of impact with the ball, which is influenced by position on the court when a stroke is made, the type of stroke played, shot direction and court position of the opponent [27]. For example, some studies suggest that match winners spend more time in the offensive zone than losers [28,29]. These aspects have been studied independently, but not through a combined analysis that could provide more effective information to improve training and subsequent performance in competition.

Based on the above, the main objective of this study is to determine the probability of winning a point in elite men's singles tennis considering several variables that the literature has shown to be key to competitive performance in tennis. Data from the late rounds of three of four Grand Slam men's singles tournaments allow the documentation of the tactical demands and success rates at the highest level of play of the sport.

In order to achieve this objective, the following hypotheses (H) will be taken into account:

H1. Most of the points are played on the first serve, with no difference between the surfaces, although there are more aces on fast surfaces.

H2. Short rallies predominate in men's professional tennis today, although they are more common on fast surfaces.

H3. Most points are won when a player hits the ball in the offensive zone, when the ball hit by the opponent bounces in the service zone or in the middle zone of the court (in zone 1 or zone 2 according to our study).

H4. On fast courts, the player on serve wins a higher percentage of points than on clay.

H5. There are combinations of variables (patterns of play) that increase the probability of winning the point.

H5a. The first serve favours the probability of winning the point, being more accentuated on fast surfaces.

H5b. The second serve significantly reduces the probability of winning the point compared to the first serve.

H5c. The first serve—short rally combination is the most likely to win the point for the server.

H5d. Point ending varies by court type, service and rally length.

## Method

### Design

In order to approach the objectives of this research, observational methodology was used [30]. The observational design [31] used is nomothetic (all points played in the final rounds of the Grand Slams in 2021 -from the quarter-finals onwards-, excluding the Australian Open), follow-up (one season), and unidimensional (there is no concurrence of behaviours).

### Sample

All points from men's matches from the quarter-finals onwards of three of the four Grand Slam tournaments of the 2021 season were registered (one per type of court surface), seven matches per tournament (1660 points at Roland Garros, 1623 at Wimbledon and 1386 at the US Open). Seventeen different men's tennis players were analysed. The study was approved by the Ethics Committee of the Faculty of Education and Sport Science (University of Vigo, application 02/0320).

### Instruments

The observational instrument for this study was made *ad hoc*, although it was based on a previously validated observation instrument on match analysis in tennis, which had similar objectives to the present research [2]. The instrument described in Table 1 is a system of categories [30] called OBSTENNIS-S21 (Tennis observational instrument for the 2021 season). The validity of the construct of the observation instrument was done by its coherence with the theoretical framework [32] and by consulting two tennis and observational methodology experts who reached a degree of agreement of 95% in response to a questionnaire about the observation instrument, analysing the suitability of it for the reality of the competition and by following the same procedure as previous studies [2,33]. The two experts were provided with a comprehensive description of the observation instrument, the objects of the investigation and instructions for answering the questionnaire. The questionnaire consisted of five items (with a Likert scale of five levels) about its suitability to the object of study, compliance with the criteria of completeness and mutual exclusivity, clarity in the wording of the categories and the degree of objectivity that allows the data collection to be unified by various observers.

The OBSTENNIS-S21 (Table 1 and Fig 1) is made up of seven criteria and 50 categories. Data registration was performed with LINCE PLUS software [34].

**Table 1. OBSTENNIS observation instrument.**

| VARIABLE | CODE | DESCRIPTION |
|---|---|---|
| SERVICE | FS | The point is played with first serve. |
| | SS | The point is played with second serve. |
| | DF | The server makes a double fault. |
| RALLY LENGHT | SH | Short rally (0–4 shots). |
| | MD | Medium rally (5–8 shots). |
| | LN | Long rally (9+ shots). |
| BOUNCE ZONE | SZ | The point ends from the service zone (ace or double fault). |
| | ZB1 to ZB5 | The zone of the court where the ball bounces before a winner or forced error (in this case the player who wins the point is registered). In the case of an unforced error, the bounce before the error is registered. In the case of a volley or smash, the area where the player's feet are placed is registered. |
| THE FINISH ZONE | Z1 to Z5 | Zone of the court where the ball is finally directed (only for winners and forced errors). |
| | NET | The final shot goes into the net or does not reach the net. |
| | LTO | The final shot goes out on the lateral side. |
| | BSO | The final shot goes out on the baseline. |
| WINNER | SW | The server wins the point. |
| | RE | The returner wins the point. |
| POINT ENDING | SWW | The server wins with a winner. |
| | SWFE | The server wins with a forced error. |
| | SWUE | The server wins with an unforced error. |
| | RWW | The returner wins with a winner. |
| | RWFE | The returner wins with a forced error. |
| | RWUE | The returner wins with an unforced error. |
| FINISH AND FINAL STROKE | SACE | The server wins with an ACE. |
| | SWFH | The server wins with a forehand winner. |
| | SWBH | The server wins with a backhand winner. |
| | SWOT | The server wins with a winner with another type of stroke (drop shot, smash, volley. . .). |
| | SFEFH | The server wins with a forehand by a forced error of the opponent. |
| | SFEBH | The server wins with a backhand by a forced error of the opponent. |
| | SFEOT | The server wins with another type of stroke by a forced error of the opponent. |
| | SUEFH | The server wins with a forehand by an unforced error of the opponent. |
| | SUEBH | The server wins with a backhand by an unforced error of the opponent. |
| | SUEOT | The server wins with another type of stroke by an unforced error of the opponent. |
| | RDF | The returner wins by double fault of the serving player |
| | RWFH | The returner wins with a forehand winner |
| | RWBH | The returner wins with a backhand winner |
| | RWOT | The returner wins with a winner with another type of stroke (drop shot, smash, volley. . .). |
| | RFEFH | The returner wins with a forehand by a forced error of the opponent. |
| | RFEBH | The returner wins with a backhand by a forced error of the opponent. |
| | RFEOT | The returner wins with another type of stroke by a forced error of the opponent. |
| | RUEFH | The returner wins with a forehand by an unforced error of the opponent. |
| | RUEBH | The returner wins with a backhand by an unforced error of the opponent. |
| | RUEOT | The returner wins with another type of stroke by an unforced error of the opponent. |

Note. A winner occurs when a player is unable to touch the ball with their racquet before it bounces twice during a match. A forced error is one where the player who commits the error is seen to have had a ball that was unreturnable. An unforced error is one where a player has a playable ball and commits a fault or hits the net with his return with no mitigating circumstances.

## Procedure

Data collection was carried out by recording the matches of three out of the four Grand Slams of the 2021 season (one on each surface). The videos were recorded at a resolution of 1080p (1920x1080). These matches were viewed for analysis on 27-inch monitors. Prior to the data quality testing, which was carried out with two experts in tennis and observational methodology, training in the use of the observation instrument was conducted. The training consisted of familiarisation with the observation tool. For this purpose, nine 2-hour sessions were held over three weeks using videos of men's tennis matches from the 2020 season.

To ensure rigour in the registration process [35], the quality of the registered data was controlled by calculating intra- and inter-observer agreement using the Kappa coefficient [36] calculated using LINCE PLUS software. Both concordances were performed on points that did not belong to the final sample (n = 450; 1/10 of the final sample). The intra-observer kappa was 0.93 for the first observer and 0.96 for the second observer. The inter-observer kappa was 0.94. After the data quality tests, observer 2 carried out the analysis of all the points in the research sample.

After registering all of the points, we obtained an Excel file with the sequence of the actions that occurred at each of the points analysed. The versatility of this Excel file allowed us to automatically transfer the information to an SPSS file, the software with which the different statistical analyses of the research were carried out.

## Data analysis

All statistical analyses were performed using IBM- Statistical Package for the Social Sciences, version 25.0 (IBM-SPSS Inc., Chicago, IL, USA). Statistical significance was assumed for $p<0.05$.

The $\chi2$ test was used to contrast the differences between the categories of each criterion used (intra-criteria analysis), as well as to compare the differences between the playing surfaces (clay, grass or hard court) of Roland Garros, Wimbledon and US Open (inter-criteria analysis).

The analysis of the probability of winning a point as a function of the combination of performance indicators selected by the researchers was carried out in three steps: firstly, we segmented the file by groups according to the court surface; secondly, a selection of cases was made (based on the serve, rally and point ending) and finally a frequency analysis with the study variable "winner" (server or returner win). To determine whether there were significant differences between the winner variable and the court surface according to a previous selection of performance indicators (based on the serve and rally), a cross-table test was carried out using the chi-square statistic.

## Results

### Analysis of variables influencing performance in men's singles tennis at Grand Slams

The information obtained from the different variables studied (Table 2) shows that, regardless of the court surface (clay, grass or hard), the majority of points started with the first serve (65.1–61.8–63.6% respectively), although there are no statistically significant differences depending on the type of court. The surface with the most aces was hard court (11.2%) and the least on clay (6%) (H1). Short rally points predominated on any type of court (64.9–77.4–68.8%) although they were more frequent on fast surfaces (H2).

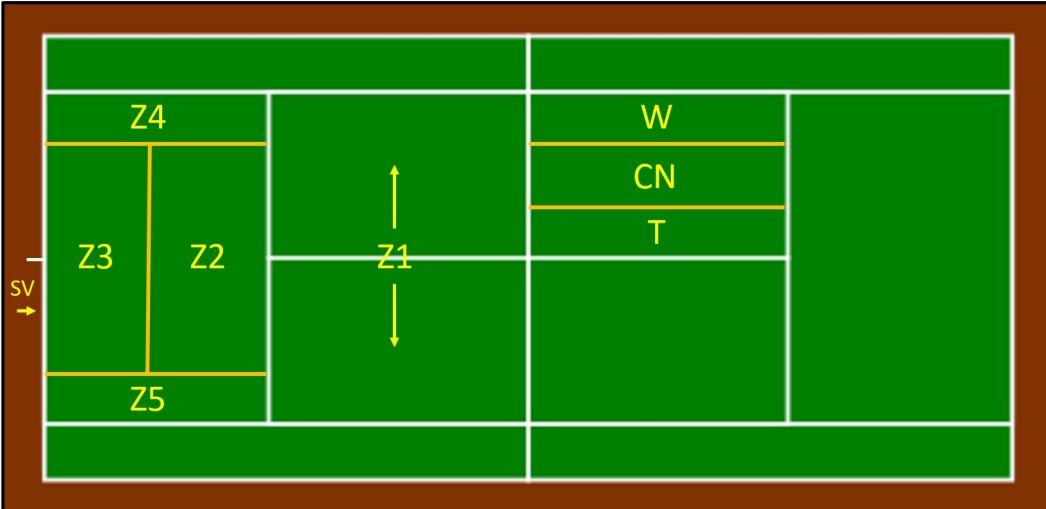

**Fig 1. Court zones.**

Nearly half of the points analysed on the three surfaces ended after a bounce previous to the final shot into zone 1 (42.8–49.1–43%) (H3) and ended with a winner or forced error in zone 1 (24.1–32.9–33.2%). The points won (with a winner or forced error by the opponent) in zone 5 also stood out, especially on clay (11.4%). The surface where most points were won on the service was grass, compared to hard court and clay (65.2–64.7.8–63%). In any case, there are no statistically significant differences (H4).

With both, serving and returning, it is on clay where more winners were obtained (24.5% and 8% respectively; on grass 22.5% and 6.2% and on hard court 23.9% and 7%), although it is also the surface (serving) where fewer points were obtained by forced error (14.8% compared to 20.6% and 18.2%). An even amount of points won by returning (23.7–24.8–24.1%) and serving (23.6–22.1–22.7%) by unforced error was registered on all three surfaces. Unforced errors on the return had similar percentages after hitting with forehand and backhand; however, when serving, they were much more frequent with the forehand. Winners were more frequent with the forehand, both serving and returning, and were considerably more frequent when serving.

Statistically significant differences (p<0.05) were found between the categories of each of the variables and on each of the three surfaces analysed (intra-variable χ2). In the comparison between surfaces (inter-variable χ2 test), statistically significant differences were observed in the type of rally length (greater number of medium and long rallies on clay than on grass and hard court), the bounce zone (greater number of actions from zone 2 and fewer from the service on clay compared to the other surfaces), the finish zone (among other aspects, fewer finishings to the net and more shots were played to the lateral zone and to zone 5 on clay compared to other surfaces), the type of point ending (on fast courts more points were won on the serve due to a forced error by the opponent) and the type of the finishing (for more details see S1 Table).

## Probability of winning a point as a function of different combinations of variables influencing performance

Fig 2 shows the server's chance of winning the point on the different court surfaces analysed as a function of the following combination of variables: service and rally length (H5).

**Table 2. Description of the performance variables of the matches on the different court surfaces in the 2021 season and comparative analysis between court surfaces (χ2 inter-variable).**

| Study variables | | Clay | | Grass | | Hard | | χ2 Inter-variable |
|---|---|---|---|---|---|---|---|---|
| | | n | % | n | % | n | % | |
| **SERVICE** | DF | 45 | 2,7 | 60 | 3,7 | 58 | 4,2 | χ2 = 8,174 |
| | FS | 1081 | 65,1 | 1003 | 61,8 | 882 | 63,6 | p = .085 |
| | SS | 534 | 32,2 | 506 | 34,5 | 446 | 32,2 | |
| **RALLY LENGHT** | LN | 223 | 13,4 | 112 | 6,9 | 182 | 13,1 | χ2 = 74,200 |
| | MD | 359 | 21,6 | 254 | 15,7 | 250 | 18,0 | p = .000 |
| | SH | 1078 | 64,9 | 1257 | 77,4 | 954 | 68,8 | |
| **BOUNCE ZONE** | SZ | 141 | 8,5 | 187 | 11,5 | 213 | 15,4 | χ2 = 66,987 |
| | ZB1 | 711 | 42,8 | 797 | 49,1 | 596 | 43,0 | p = .000 |
| | ZB2 | 441 | 26,6 | 339 | 20,9 | 313 | 22,6 | |
| | ZB3 | 197 | 11,9 | 186 | 11,5 | 164 | 11,8 | |
| | ZB4 | 99 | 6,0 | 56 | 3,5 | 57 | 4,1 | |
| | ZB5 | 71 | 4,3 | 58 | 3,6 | 43 | 3,1 | |
| **THE FINISH ZONE** | BSO | 278 | 16,7 | 234 | 14,4 | 218 | 15,7 | χ2 = 87,508 |
| | LTO | 229 | 13,8 | 155 | 9,6 | 121 | 8,7 | p = .000 |
| | NET | 279 | 16,8 | 332 | 20,5 | 269 | 19,4 | |
| | Z1 | 400 | 24,1 | 534 | 32,9 | 460 | 33,2 | |
| | Z2 | 94 | 5,7 | 103 | 6,3 | 81 | 5,8 | |
| | Z3 | 56 | 3,4 | 42 | 2,6 | 34 | 2,5 | |
| | Z4 | 135 | 8,1 | 97 | 6,0 | 105 | 7,6 | |
| | Z5 | 189 | 11,4 | 126 | 7,8 | 98 | 7,1 | |
| **WINNER** | RW | 615 | 37,0 | 565 | 34,8 | 489 | 35,3 | χ2 = 1,972 |
| | SW | 1045 | 63,0 | 1058 | 65,2 | 897 | 64,7 | p = .373 |
| **POINT ENDING** | RWFE | 89 | 5,4 | 62 | 3,8 | 58 | 4,2 | χ2 = 26,960 |
| | RWUE | 393 | 23,7 | 402 | 24,8 | 334 | 24,1 | p = .003 |
| | RWW | 133 | 8,0 | 101 | 6,2 | 97 | 7,0 | |
| | SWFE | 246 | 14,8 | 335 | 20,6 | 252 | 18,2 | |
| | SWUE | 392 | 23,6 | 358 | 22,1 | 314 | 22,7 | |
| | SWW | 407 | 24,5 | 365 | 22,5 | 331 | 23,9 | |
| **FINISH AND FINAL STROKE** | RDF | 38 | 2,3 | 60 | 3,7 | 58 | 4,2 | χ2 = 172,046 |
| | RFEBH | 23 | 1,4 | 22 | 1,4 | 25 | 1,8 | p = .000 |
| | RFEFH | 61 | 3,7 | 30 | 1,8 | 29 | 2,1 | |
| | RFEOT | 5 | ,3 | 10 | 0,6 | 5 | 0,4 | |
| | RUEBH | 133 | 8,0 | 77 | 4,7 | 105 | 7,6 | |
| | RUEFH | 183 | 11,0 | 196 | 12,1 | 150 | 10,8 | |
| | RUEOT | 40 | 2,4 | 69 | 4,3 | 20 | 1,4 | |
| | RWBH | 28 | 1,7 | 24 | 1,5 | 26 | 1,9 | |
| | RWFH | 73 | 4,4 | 42 | 2,6 | 50 | 3,6 | |
| | RWOT | 31 | 1,9 | 35 | 2,2 | 21 | 1,5 | |
| | SACE | 99 | 6,0 | 125 | 7,7 | 155 | 11,2 | |
| | SFEBH | 92 | 5,5 | 152 | 9,4 | 101 | 7,3 | |
| | SFEFH | 139 | 8,4 | 166 | 10,2 | 137 | 9,9 | |
| | SFEOT | 16 | 1,0 | 17 | 1,0 | 14 | 1,0 | |
| | SUEBH | 172 | 10,4 | 135 | 8,3 | 145 | 10,5 | |
| | SUEFH | 193 | 11,6 | 181 | 11,2 | 157 | 11,3 | |
| | SUEOT | 26 | 1,6 | 42 | 2,6 | 12 | 0,9 | |
| | SWBH | 51 | 3,1 | 22 | 1,4 | 28 | 2,0 | |
| | SWFH | 168 | 10,1 | 112 | 6,9 | 89 | 6,4 | |
| | SWOT | 89 | 5,4 | 106 | 6,5 | 59 | 4,3 | |

*Note.* Abbreviations in Table 1.

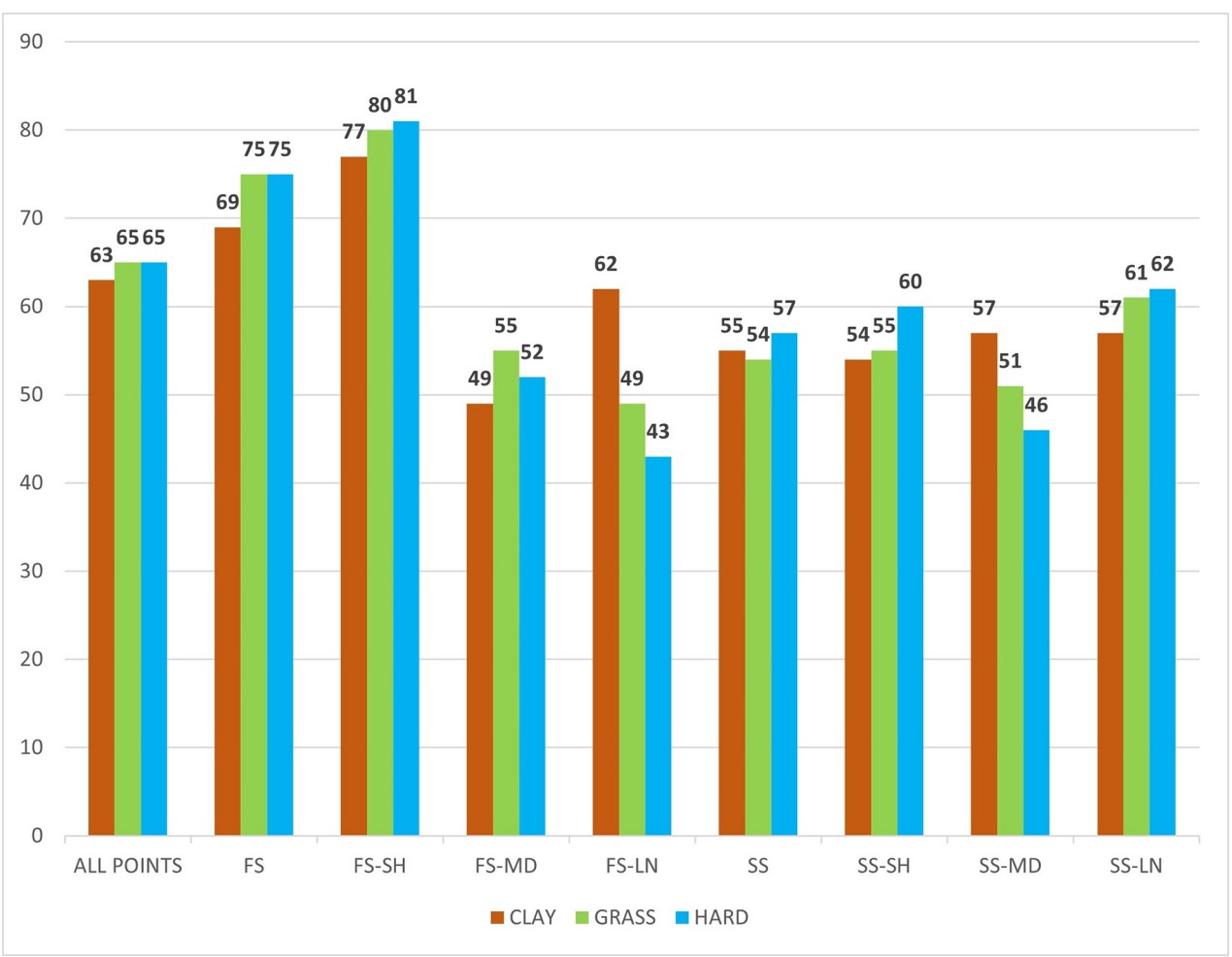

**Fig 2. Probability of winning the point on each surface as a function of type of service and rally length.**

The probability of winning a point with the first serve was lower on clay than on grass or hard court (69-75-75% respectively) (H5a). The chance of winning the point with the second serve was very similar on all three surfaces (55-54-57%), which meant a decrease in the probability on all surfaces relative to the first serve (reduction of 14-21-18% respectively) (H5b). The first serve—short rally combination was the most successful for the server in terms of winning points (77-80-81%) (H5c).

Statistically significant differences were observed between court surfaces in the combination first service-long rally ($\chi 2 = 8.052$; sig. = 0.018). On both, grass and hard court, the returner was more successful in this type of combination, but not on clay.

Figs 3 and 4 show an analysis of point endings (winners, forced errors or unforced errors) on the different court surfaces based on the following combination of variables: service and rally length (H5d).

Significant differences between court surfaces were found in the first service–short rally combination ($\chi 2 = 8.052$; sig. = 0.018) and in the first service–long rally combination ($\chi 2 = 26.106$; sig. = 0.004). In the first of the aforementioned combination, a greater number of points ended by unforced errors were registered on clay than on the rest of the surfaces, both on the serve and on the return. Similarly, the number of forced errors (made by the returner)

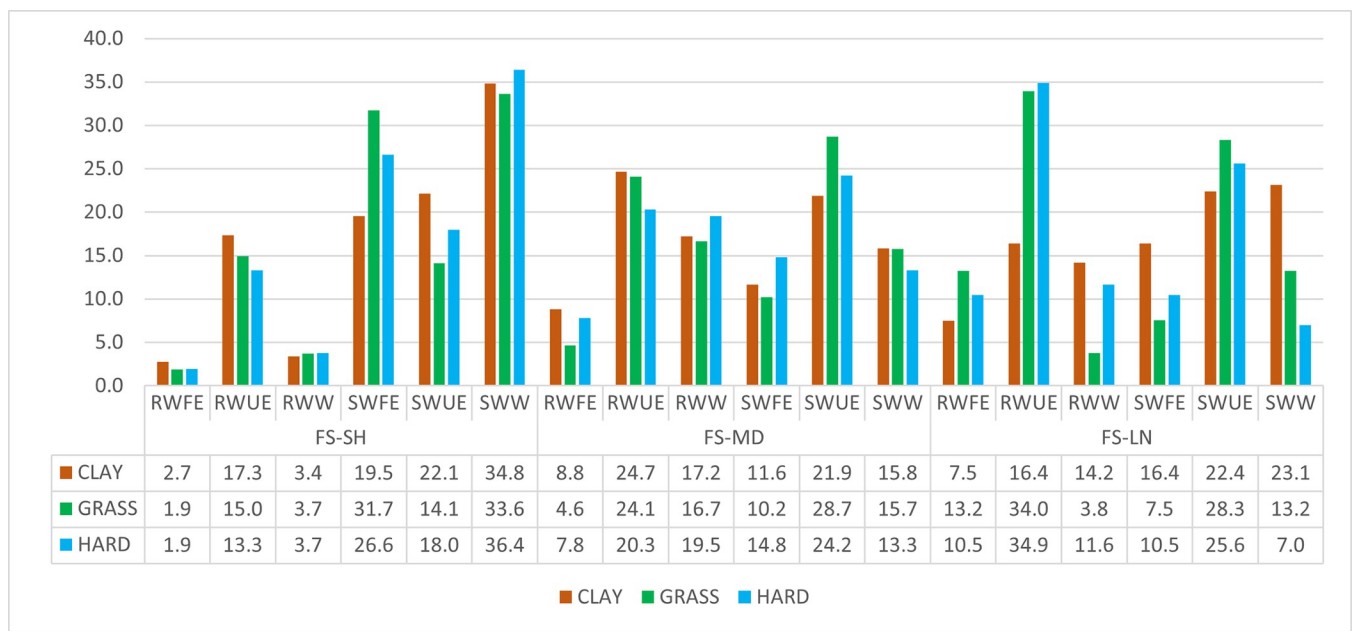

**Fig 3. Point ending with first service.**

won when serving was higher on grass than on the other surfaces. In the second combination, the number of unforced errors in points won on the return was far higher on grass and hard than on clay. There were also more winners on clay than on grass and hard (in both serving and returning points).

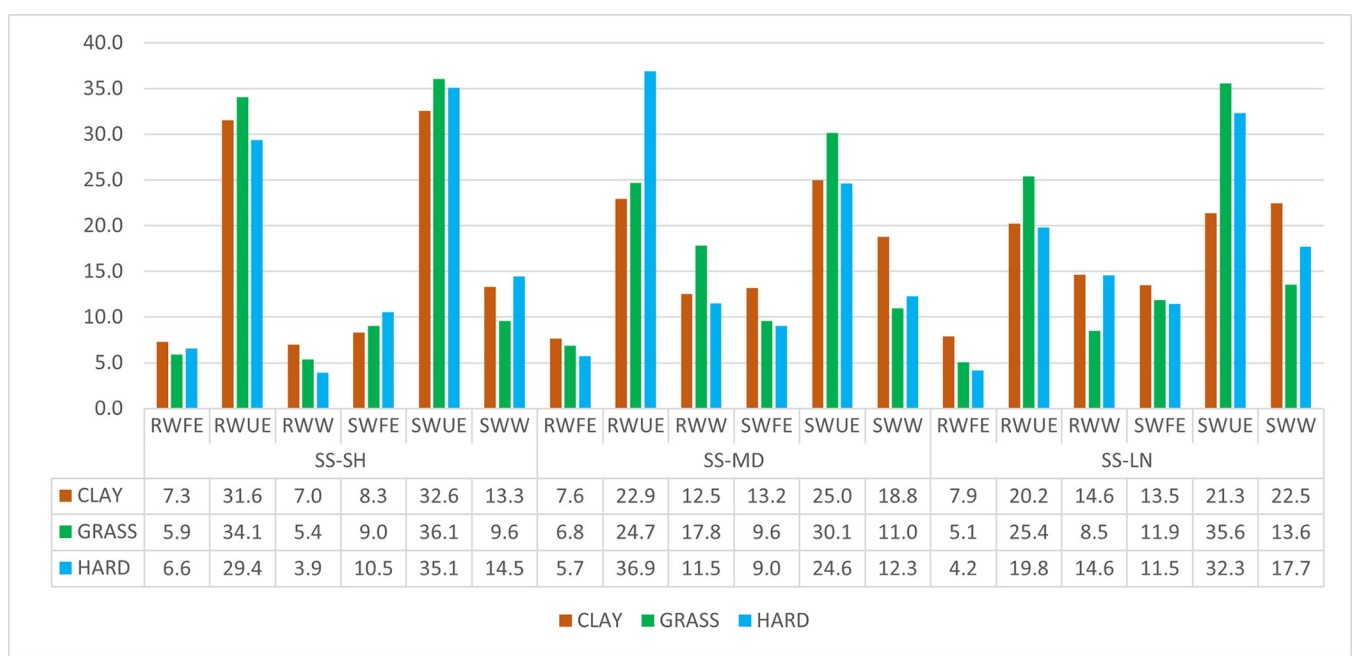

**Fig 4. Point ending with second service.**

## Further statistical information of variables influencing performance

S2 Table shows the points observed as a function of the surface considering the following variables: service (first and second) rally length (short, medium and long), bounce zone (ZB1-ZB5), the finish zone (Z1-Z5 or BS, LT and NT) and point ending (server or returner won the point by forced error, unforced error or winner) (H5).

The most frequent combination based on the serve, rally length and bounce zone found was first serve, short rally and final shot after a bounce in zone 1, representing 25-35-30% of the total depending on the surface (clay, grass and hard court). In this situation, only 14% of the points were won on the return (the value is the same on all three surfaces). When serving, it is on clay where more points were won by winners (23%) and by unforced errors by the opponent (33%). The percentage of winners to zone 1 (41%) and zone 5 (28%) stands out in comparison with the others. Unforced errors are preceded by a shot that goes out over the baseline (zone 3–52%-). On grass, forced errors (48%) were more frequent and were mostly directed to zone 1 (88%). This was also the case with winners (52%), but those directed to zone 2 (15%) and zone 5 (16%) also represent a significant proportion of the total. The points won by serving through an unforced error by the opponent are usually sent to the net (59%). Most of the points won by the returner are by unforced error with a shot to the net (59%). On hard court, the data were similar to that of grass, with a high number of forced errors (46%) played to zone 1 (93%). Winners (16%) were mostly directed to zone 1 (44%) but there were shots played to all other zones (around 15%).

S3 Table shows an analysis of the points registered as a function of the surface considering the following performance indicators: service (first and second), rally length (short, medium and long), finish (forced error, unforced error and winner) and final stroke (backhand, forehand or other strokes).

On clay, winners, when the player served, were more frequent with the forehand than with the backhand in all the possible match situations studied, something that did not occur on grass or hard court. Returning, winners were predominantly with the forehand on clay and hard court, but on grass, many points were resolved with another type of stroke (probably volleys). Unforced error points won on the serve were very even on clay with first serves (regardless of the type of rally) and second serves in short rallies. On grass and hard courts, this balance was only evident with first serves in short and medium rallies. In all other cases, forehand errors predominated, except in short rallies in second serves on hard courts.

The points won by unforced errors on grass, when the player was returning, was linked to a forehand stroke (occurred in all possible solutions). On clay it was the same in short rallies (both first and second serves), there was a balance between forehand and backhand errors in medium rallies and in long rallies the error was preceded by a backhand stroke. On the hard surface, there was a balance of forehand and backhand errors in almost all the situations analysed, except for the first serve and short rallies, where the errors were clearly caused by a forehand stroke.

## Discussion

This study identifies several patterns (combined information about different variables that influence performance) that determine the probability of winning a point in elite men's singles tennis, which can help players and coaches define match strategy.

The results obtained on first serve accuracy (ball put into play) are similar to the results found in other research on elite men's singles tennis [3,18] (58–64% on clay, 65% on grass and 63–65% on hard court), which confirms H1. It was also evident that aces were more common on hard surfaces (grass and hard court) than on clay [10,20], which supports H1. Most of the

points, although there are differences between surfaces, were played with a short rally (64.9% on clay, 77.4% on grass and 68.8% on hard), results that confirm H2. The literature has contrasted that, for many years, the frequency of this type of rally has increased on all surfaces due to the increase in speed in tennis, even though for a period of time there was a slowdown due to the change in the type of balls (type III balls) [37]. This increase in speed is determined by various factors, such as improvements in racquet designs [38,39], greater physical ability or more aggressive playing strategies by players, where they put pressure on the opponent from the start [19,40,41].

As expected, most of the points are ended after a player hits a ball that bounces in the service zone or in the middle zone of the court (zone 1 or zone 2), an aspect that confirms the proposed H3. Several studies indicate that playing in the offensive zone of the court increases the chances of winning the match [42]. It has also been confirmed that the winners/forced errors are mainly directed to zone 1 (we assume that they are looking for angles that are difficult to return) and also to the opponent's backhand zone (zone 5).

The data from this study does not fully confirm H4. Taking all points analysed, the server won 63% of all points on clay and 65% on grass and hard, so the differences are small. Previous studies have shown larger differences in the number of points won on different surfaces [20], which could be seen as evidence that tennis is evolving and that it is important to constantly analyse the variables that affect performance. The differences are clearer if we consider only points played with first serves (69% of points won on clay and 75% on grass and hard courts) or if we compare first serves combined with short rallies. Clay courts in Roland Garros induces slower and higher ball bounce, providing the receiver with the opportunity of returning more serves than on faster surfaces such as grass in Wimbledon [26].

The probability of winning a point when a point is initiated with a first serve was similar to that contemplated by other researchers in the same context [3,18,23] (69–75% on clay, 75–79% on grass and 70–78% on hard court). According to the researchers cited above, the chance of winning a point with a second serve fluctuates depending on the surface (between 51–56%, 53–58% and 48–55% respectively), values also close to that obtained, in our case, in the 2021 season. These results are consistent with the friction coefficient and coefficient of restitution data that place grass as the surface with the highest speed, ahead of hard court and clay [43]. It is confirmed that the first serve is fundamental to win matches in this sport, especially on fast surfaces [15,16,21,44] as the loss of efficiency with second serves is notable, being more accentuated on fast courts. Considering that playing second serves generates a probability of winning the point of around 55%, first serves might be an effective match strategy. Current data suggests that for elite men players, starting the point with a first serve instead of a second serve can increase the probability of winning the point by 14–21%, depending on the surface. This information corroborates H4a and H4b.

The study confirmed that the combination of first serve—short rally is the most likely to win the point for the server, something that has already been pointed out by several recent studies and which confirms H4c [15,45,46]. It has also been shown that players who dominated short rallies (0–4 shots) won the match in 9 out of 10 cases on both clay and grass [17,25], so tactical training should try to improve this aspect of the game.

The best strategy for the player that was serving with a first serve and short rally (server won between 77–81% of all points) was to achieve a winner, as 34–36% of the cases obtained the point later on either surface (taking into account the ending of all points). In addition, especially on grass, a high percentage of forced errors by the opponent (points won by the server) was registered (32%), dropping to 27% on hard and 20% on clay (data refers to all points). The best return strategy was to seek an unforced error from the opponent, regardless of the surface, although this was more effective on clay (17%) than on grass (15%) or hard

court (13%). Return winners after the first serve were rather scarce in short rallies (3–4%). With this information, and considering that the returner is probably always at a tactical disadvantage [16], training to work on the defence of the serve should be aimed at trying to put the server in a situation of discomfort, rather than trying to find a winner, at least until after four shots. Hitting the ball to zone 3 (baseline) and zone 4 (backhand on the right-handed player) were the most effective, confirming that the search for the opponent's error occurs more frequently on the backhand stroke [27].

Medium rally points represented 21.6–15.7–18.0% of all points in this study depending on the surface. According to one study [25], dominance on these points determines the winner of a clay court match by 65% and 69% on grass (approximately 25% less than dominance on short rallies). With the first serve, the tactic of looking for a winner on either surface was no more effective than seeking an unforced error from the opponent. It was the same for the returning player. The points won by unforced error were very balanced between server and returner, so it seems clear that, at a strategic level, the server has no interest in reaching this type of rally [45]. This highlights the importance of having an effective service to end the point as soon as possible and thus improve the probability of success at the point [15,16,45,47]. With the second serve, the statistics do not vary much with respect to the first serve. In fact, although on fast surfaces and medium rallies there was a decrease in the probability of winning the point between the first and second serve (from 55% to 51% on grass and from 52% to 46% on hard courts, on clay there was even an increase from 49% to 57%). This would suggest that the effect of the serve is diluted after the fifth shot, something that has already been suggested in the literature [45] and which again corroborates the H5c. On clay, with this rally and second serve, the tactic of looking for a winner (winner or forced error) was more effective than looking for the unforced error. In the return game, the opposite was true. On grass it was more effective to look for the unforced error both on the serve and on the return. On hard courts it was much more effective to look for the unforced error, especially in the return.

Long rally points were few in general, although this fact was more pronounced on grass, where not even 7% were observed. Although it was not investigated whether these points were important points (e.g.: breakpoints), the literature has pointed out that dominance on these types of points is linked to match success in 66% of cases on clay and 61% on grass [17], quite similar to that of medium rallies, so they must also have their importance in training, both from a physical, psychological and strategic point of view. In fact, there is a curiosity, at least on fast surfaces. The server won fewer points after a first serve than the returner, but significantly increased his effectiveness with a second serve. The explanation for this phenomenon is complex, but it may be possibly linked to aspects of concentration, due to starting from a disadvantageous situation (second serve). Although this data provides information on the probability of winning the point with a long rally, it would not make sense to make a recommendation to a player who is going to serve on the basis of the information provided.

Considering the information described in these last three paragraphs (each of which focused on one type of rally), and where different combinations of variables affecting the probability of winning the point were carried out, we can confirm H5 of the study, which supports that there are patterns of play that should be taken into account for improving match strategy in elite men's singles tennis competition.

## Conclusions

The first serve in elite men's singles tennis is essential to increase the chance of winning the point on all types of rally and surfaces. Starting the point on the second serve decreases the chances of winning the point by 14.6% (clay), 21% (grass) or 17% (hard court) compared to

the first serve (success rate: 69-75-75% respectively). The probability of winning the point with a first serve and medium rally decreases by 25–29% depending on the surface compared to finishing with a short rally (success rate: 77-80-81% respectively). On grass only 7% of points end with a long rally (13% on the other two surfaces). The most common point winning combination of shot for the server was the one in which the point started with a first serve, there was a rally of less than five shots and the point ended after a bounce of the ball in the service zone (zone 1) and a subsequent aggressive shot. This generated an unforced error in the opponent (on clay) or a forced error (on grass and hard court).

## Practical application

Throughout this article, although focusing on service, surface and rally length, different information has been established for the probability of winning a point depending on different variables. The stroke patterns and success/errors rates are what are currently present at the highest levels. Male players seeking this level of play should consider training and shots in match play that align with greater probability of success. We recommend that coaches make use of the different tables and figures in the manuscript, as well as the supplementary material to obtain more specific data, where an analysis is also made according to the bounce zone prior to the last shot, type of final stroke and direction of the final shot, all differentiated by surface, type of service and rally length.

## Limitations and future perspectives

In this study we have only analysed matches from the Grand Slam quarter-finals onwards, so previous rounds have not been taken into account. Therefore, only a small number of players were analysed who, in addition, could be under the effects of psychological stress and physical fatigue. The stroke patterns and error rates observed may not be similar for players from other sex, skill level, or disabilities. The direction of the serve (wide, body or T zone) and its speed have not been considered, aspects that could be interesting to address in future research.

Bearing in mind that most of the points end in short rallies (0–4 shots), in the future studies should be carried out linked to the exact number of shots in the point, focusing on this type of rally, in order to establish more specific point winning probabilities, taking into account all the movements and strokes made by the players.

## Supporting information

**S1 Table. Description of variables that affect performance on the different court surfaces in the 2021 season, analysis of the distribution of categories of each variable by surface (intra-variable $\chi$2) and comparative analysis between court surfaces ($\chi$2 inter-variable).** (DOCX)

**S2 Table. Analysis of different combinations of variables that affect performance (service, rally length, bounce zone, the finish zone and point ending) as a function of the court surface.** (DOCX)

**S3 Table. Analysis of different combinations of variables that affect performance (service, rally length, final stroke and point ending) as a function of the court surface.** (DOCX)

## Author Contributions

**Conceptualization:** Iván Prieto-Lage, Adrián Paramés-González, Daniel Torres-Santos, Juan Carlos Argibay-González, Xoana Reguera-López-de-la-Osa.

**Data curation:** Adrián Paramés-González, Juan Carlos Argibay-González, Xoana Reguera-López-de-la-Osa.

**Investigation:** Iván Prieto-Lage, Adrián Paramés-González, Daniel Torres-Santos, Xoana Reguera-López-de-la-Osa.

**Methodology:** Iván Prieto-Lage, Daniel Torres-Santos, Alfonso Gutiérrez-Santiago.

**Project administration:** Adrián Paramés-González.

**Resources:** Iván Prieto-Lage, Daniel Torres-Santos, Juan Carlos Argibay-González.

**Software:** Iván Prieto-Lage, Juan Carlos Argibay-González.

**Supervision:** Iván Prieto-Lage, Alfonso Gutiérrez-Santiago.

**Visualization:** Daniel Torres-Santos, Juan Carlos Argibay-González.

**Writing – original draft:** Iván Prieto-Lage, Daniel Torres-Santos, Xoana Reguera-López-de-la-Osa, Alfonso Gutiérrez-Santiago.

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
