## [Decision Letter · Decision Letter 0]

23 Jan 2023

PONE-D-22-33800Analysis of effectiveness through patterns of play in professional men's tennisPLOS ONE

Dear Dr. Prieto-Lage,

Thank you for submitting your manuscript to PLOS ONE. After careful consideration, we feel that it has merit but does not fully meet PLOS ONE’s publication criteria as it currently stands. Therefore, we invite you to submit a revised version of the manuscript that addresses the points raised during the review process.

ACADEMIC EDITOR:

Dear Authors,

Two experts in the field revised your current manuscript and recognised some important points that should be addressed.

We look forward to receiving your revised manuscript.

Kind regards,

Javier Abián-Vicén, Ph.D.

Academic Editor

PLOS ONE

Journal Requirements:

Reviewers' comments:

Reviewer's Responses to Questions

**Comments to the Author**

1. Is the manuscript technically sound, and do the data support the conclusions?

Reviewer #1: Yes

Reviewer #2: Partly

2. Has the statistical analysis been performed appropriately and rigorously? 

Reviewer #1: Yes

Reviewer #2: No

3. Have the authors made all data underlying the findings in their manuscript fully available?

Reviewer #1: Yes

Reviewer #2: Yes

4. Is the manuscript presented in an intelligible fashion and written in standard English?

Reviewer #1: Yes

Reviewer #2: Yes

5. Review Comments to the Author

Reviewer #1: Initial comments

The reviewer would like to congratulate the authors for conducting research in tennis. More specifically, for studying the strategy of the game at the high-performance level.

The purpose of this review is to share with the authors several comments that will help to improve the quality of the manuscript since, in the views of the reviewer, there are some relevant aspects that need to be clarified, changed, referenced and deleted.

To assist in the review, the reviewer has enclosed the original pdf of the manuscript with his suggestions and comments. By doing this, the authors will find it easier to follow the recommendations.

Structure of the paper

No comments regarding the structure of the manuscript.

Language

The reviewer suggests that the language is double checked by an English native expert. More importantly, it has to be someone that is familiar with the game of tennis, since some of the words and constructions used in the text are not completely accurate.

Introduction

The authors mention the concept of patterns of play but this concept is not defined in the manuscript. Throughout the document there seems to be a confusion between shots, patterns, behaviours, sequences, etc. This has to be clarified by the authors using the adequate references.

The authors state the objective of the paper but they do not include any hypothesis that they would like to test. This is a very important aspect that also needs to be addressed. Hypotheses have to be included and then, the results and the discussion should refer to them.

Methodology

The authors use an observational methodology based on an instrument that it is not indicated if it has been used in previous research nor if it has been validated. Again, this has to be clarified to ensure that the tool used is adequate for the purposes of the study.

Furthermore, they use a division of the tennis court in zones that, again, it is not indicated if it has been used in previous studies, nor if it has been validated. The reviewer is not sure where they have taken the references to divide the court in those zones.

Finally, with the technological advances available nowadays, the notational analysis is mostly conducted using other types of methodologies provided by the statistics and computer programmes available.

Results and discussion

The reviewer feels that the results should be structured based around the hypotheses set. If this is not the case, they will not be clear enough.

Conclusion

They seem adequate for the goal of the paper. However, the reviewer feels that the limitations and the practical applications should be part of the conclusion and not be included in the discussion.

Final recommendation

Due to the comments made above, the reviewer feels that the manuscript needs some major changes that will substantially improve its quality.

Finally, the reviewer would like to thank again the authors for their passion and effort in producing this manuscript in tennis.

Miguel Crespo, PhD.

Reviewer #2: The report provides a retrospective analysis of categorical data of elite male tennis players in the final rounds of the most prestigious tournaments. The report suffers from unclear writing that makes it difficult for even motivated, tennis-interested readers to follow. The specific comments to the authors outline weaknesses that should be addressed.

Page-Line

2-4 . . . understanding of tactical actions in tennis. In particular, sequential analysis is needed to understand the relationships between actions in competitions.

2-8 Add wording that all matches at these tournaments were analyzed and if these represent males, females, or both

2-10 Clarify wording if these are means and what is “effectiveness?” Ball put in play or point won?

2-14 . . . won by the serving player? Medium rallies or longer?

2-17 Revise. You have no prospective evidence of tennis training. You have descriptive data of typical patterns of play in elite tennis players. This only informs what are typical and likely effective tactics and could inform coach and player judgments on training and match strategy

3-3 Revise this incorrect sentence. Like many sports, tennis has a long history of qualitative and quantitative performance analysis—most predating the automatic data collection systems seen recently in modern sport. This point is well presented in the Reid et al (5) article you cite, although they ignore classic texts in tennis (Classic textbooks by Talbert & Old, 1956, 1962, 1968, 1977 on singles and double tennis), early but influential tennis notation systems like COMPUTENNIS (Daw & Burton 1994; Kahn et al. 2004) and iCODA (O’Donoghue 2014; Wright et al. 2012). Your paper also cites quite a few more recent studies of notational data on tennis. The key observation here is that automated data collection and reporting systems has just accelerated recently because of these tools and growing sport/coaching awareness of the value of analytics. The first issue of the Journal of Sports Analytics was only 2015 and Int J Perform Analy Sport in 2001

3-11 to 13 Revise to clarify. Last sentence is good but next to last is an exaggeration/hyperbole. It is rare that any set of numbers can “optimize” training or performance. Even if such a complex system in a complex sport could be somewhat likely enhanced, in professional tennis there are limits on what a coach can communicate with players during a match

3-14 Revise this hyperbole: Delete implied and unobtainable optimality and focus on the search for meaningful variables for performance enhancement

3-22 Please stop with the meaningless hyperbole and one-sentence paragraph? What is optimal results in tennis? Winning every point in the match? Clearly communicate previous results and their meaning. How can a reader believe you that which stroke in tennis in any situation can be perfectly predicted, although it depends on many other things, and “in any case” the sport is just about the serve?!

4-10 Vague wording. Different tournaments have different surfaces AND natural surfaces (clay & grass) have within tournament variation over time

4-13 Revise for clarity. You are referring to the position on the court when a stroke is made . . . .shot direction and court position of the opponent

4-18 Reword for clarity. Do you mean the chance of winning a point in a tennis match based on what or number of performance indicators?

5-6 Analysis of only final rounds does not agree with wording in the abstract

5-11 Again, note if these are matches with males, females, or both. Also report the sample size since some players win and compete again, as well as may be in several tournament late rounds

6-Table 1 Correct tennis terminology (e.g., double fault). Perhaps have a table note with operational definitions of common tennis terms that may not be obvious to readers (e.g., forced versus unforced error)

7-4 Describe extent of training. Size and resolution of video, television used, and ability of observers to replay video sequences

7-11 Vague. Please expand on what transformations were made and how this relates to the results of this previous study. Why are the results from reference 27 not summarized in the introduction (page 3)?

7-17 It is unclear how your rationale of a sequential analysis for this study aligns with your comparative analysis of tennis performance categorical data. Why is p < 0.05 is appropriate for this study given you have numerous Chi squared tests of likely correlated dependent variables. The family-wise type I error rate for this study is quite large, so you likely have some effects falsely identified as significant and you don’t know which ones are. An effective and easy approach is the progressive Holm (1979) correction

7-last line Describe this technique or cite a reference so readers can replicate your study

8-3 Good scientific reporting will have a narrative describing key results and parenthetically refer to the numerous specifics in a table. Combine this revised sentence with your results paragraphs

9-3 Is this correct? Inter-criterion for service was p = 0.085 and winners p = 0.373

9-11 This paragraph mixes vague observations of nonsignificant and statistically significant

10-9 Correct same error as line 8-3. Describe results and only parenthetically refer details in figure. Figure 2 needs a clear caption to facilitate reader understanding. This is also hard to understand given the vague presentation of what is meant by “effectiveness” and “sequential.” What is RG, WI, US?

11-10 through page 12 Consider condensing with only general results for interested readers to explore. The problem with you data are that they are likely biased by the handedness and players in these late rounds. It is hard to make sense of all these trends and what they might mean given the interaction with other factors (player, environment, match situation, handedness, etc)

13-4 Readers should not be finding out in the discussion that the data only refer to male professional players

13-6 Do not repeat your reporting of means by surface. The report improved by integrating this idea with your reporting of previous values

13-19 . . . first serves might be an effective match strategy. The current data indicate that this might increase points early between 14 and 21% for elite players depending on the surface

13- last line Do not refer just the last few years. The speed of tennis has been increasing for many years due to improvements in racket design (Haake & Brody 1999; Haake et al. 2000, 2007) and was even considered slowing by changes (type III ball)

14-5 Unclear. What does it mean to look for a winner?

14-8 What does it mean to look for an unforced error?

16-5 Add additional limitation of small number of players and potential interaction on fatigue/stress of additional rounds of winners that are apparently treated as independent matches/observations and are not. Weakness of inflation of type I errors also needs to be noted if not controlled

Conclusions: Nice summary. Perhaps reword to chance of winning a point rather than “effectiveness” and other wording for a “final shot” (aggressive/challenging second shot)

6. PLOS authors have the option to publish the peer review history of their article (what does this mean?). If published, this will include your full peer review and any attached files.

Reviewer #1: **Yes: **Miguel Crespo

Reviewer #2: No

---

## [Author Response · Author response to Decision Letter 0]

8 Mar 2023

A document with the response to the reviewers is attached.

---

## [Decision Letter · Decision Letter 1]

30 Mar 2023

PONE-D-22-33800R1Match analysis and probability of winning a point in men's tennisPLOS ONE

Dear Dr. Prieto-Lage,

Thank you for submitting your manuscript to PLOS ONE. After careful consideration, we feel that it has merit but does not fully meet PLOS ONE’s publication criteria as it currently stands. Therefore, we invite you to submit a revised version of the manuscript that addresses the points raised during the review process.

ACADEMIC EDITOR:I have completed my evaluation of your manuscript. I would like to thank the authors for their efforts on this second version of the manuscript. The reviewers recommend reconsideration of your manuscript following major revision. I invite you to resubmit your manuscript after addressing the reviewers' comments. 

We look forward to receiving your revised manuscript.

Kind regards,

Javier Abián-Vicén, Ph.D.

Academic Editor

PLOS ONE

Additional Editor Comments:

I have completed my evaluation of your manuscript. I would like to thank the authors for their efforts on this second version of the manuscript. The reviewers recommend reconsideration of your manuscript following major revision. I invite you to resubmit your manuscript after addressing the reviewers' comments.

Reviewers' comments:

Reviewer's Responses to Questions

**Comments to the Author**

1. If the authors have adequately addressed your comments raised in a previous round of review and you feel that this manuscript is now acceptable for publication, you may indicate that here to bypass the “Comments to the Author” section, enter your conflict of interest statement in the “Confidential to Editor” section, and submit your "Accept" recommendation.

Reviewer #1: All comments have been addressed

Reviewer #2: (No Response)

2. Is the manuscript technically sound, and do the data support the conclusions?

Reviewer #1: Partly

Reviewer #2: Partly

3. Has the statistical analysis been performed appropriately and rigorously? 

Reviewer #1: Yes

Reviewer #2: Yes

4. Have the authors made all data underlying the findings in their manuscript fully available?

Reviewer #1: Yes

Reviewer #2: Yes

5. Is the manuscript presented in an intelligible fashion and written in standard English?

Reviewer #1: Yes

Reviewer #2: Yes

6. Review Comments to the Author

Reviewer #1: Match analysis and probability of winning a point in men's tennis.

2nd review

Reviewer: Miguel Crespo, International Tennis Federation

Introduction

The reviewer would like to thank the authors for their efforts on this second version of the manuscript. In the reviewer's opinion, so many changes have been made that it really does look like a new document.

In addition, the reviewer acknowledges that the authors have considered many of the suggestions made in the first revision.

Similarly, the reviewer has noted that the authors have also taken into account many of the suggestions made by the second reviewer.

It is also noted that the authors have made considerable improvements in the English language, which makes the manuscript easier to understand.

However, despite the improvements made and the fact that the changes are so substantial that a new text could be considered, this reviewer considers that, without detracting from the interest of the work, there are still aspects that could be clarified and improved.

Therefore, in this second revision, the reviewer proposes to continue the academic debate raised by this new version in order to improve the manuscript as much as possible.

To this end, the following paragraphs are intended to assist in the improvement and development of the text presented.

Abstract

The authors have changed the abstract considerably but, in the reviewer's opinion, there are still some aspects that are unclear and contribute to a general confusion that does little to aid understanding of the text.

For example, they start by talking about tactical actions in tennis and then include the term technical-tactical performance factors. They then discuss the most important performance indicators in the sport. Obviously, the mention of three very similar terms, all related to each other, does not help to clearly understand what the authors intend. It seems as if they want to follow the well-known phrase: "If you can't convince them, confuse them".

And that is what happens to this reviewer with this new version in general. A constant confusion that, added to the lack of objectives and hypotheses, produces a sense of unfamiliarity and general lack of precision that does not help the manuscript.

In some sentences, such as "The end of the point (winner, forced error and unforced error) varies according to the court surface, the type of serve and the length of the rally" the obviousness is clear and probably does not need to be included.

Moreover, the final sentence is so ambiguous and general that it needs to be more specific to give practical value to the results of the study.

Introduction - KPIs

The changes made in the first part of the introduction, some of them suggested by the other reviewer, are appropriate and improve the quality of the previous version.

However, having reread this introduction and, in particular, the second part where the authors should go into detail about the research carried out on the subject of their study, this reviewer notes a lack of depth in the reference to the results of this research.

And this is where the reviewer notes a general confusion regarding the concept of key performance indicator. In this case, it seems that the authors consider serve and court surface to be key performance indicators in the same way as rally duration.

With all due respect, this reviewer believes that this is where the major confusion in this study lies. The serve, like the rest, the forehand or any other stroke, are technical actions or gestures. They can in no way be considered key performance indicators.

Similarly, the court surface, as well as the height of the court (at sea level or at altitude), or whether it is an indoor or outdoor court, or even the type of balls with which the match is played, cannot be considered as key performance indicators.

The fact that the authors have eliminated the different terms used to refer to what they call study variables (patterns, combinations of blows, behaviours, etc.) does not mean that the terminological confusion is not present throughout the text and should therefore be clarified and suitably modified to prevent the study, its approach, results and conclusions from being unclear and imprecise.

Similarly, when they refer to combinations of strokes (the so-called patterns), as in the case of the serve + a stroke, for example, they are not key performance indicators either, but rather patterns of play (obviously tactical) made up of combinations of gestures, but not KPIs.

Objective and hypotheses

With regard to the lack of hypotheses in the study, the authors justify this fact by the type of study in question. However, once again, in the opinion of this reviewer, if the authors do not wish to include hypotheses, which is regrettable considering the large number of previously conducted studies to which they could refer and from which to obtain possible hypotheses to validate in their research, they should at least include more objectives that would help them to structure the results and discussion section much better.

In this sense, the reviewer considers, once again, that, lacking clear, specific hypotheses based on the results of previous studies, which are not taken as a basis for this observational study, the research we reviewed lacks reference points as only one objective is established.

Again, with respect, the reviewer feels that in light of the results of previous research, the authors should set many more objectives to inform their study. In fact, a mention to the results of these previous studies should be included in the last section of the introduction to provide the necessary background to the study and to show that the authors have done the adequate homework in their literature study by extracting the main results of these previous works.

As they indicate in the new version “Based on the above, the objectives of the research were to analyse the technical-tactical performance indicators of men's Grand Slam tennis matches, as well as to study the probability of winning a point”, if these are the two objectives of the study, they should be stated more specifically based on the results of research previously obtained.

Methodology – instruments

It is important to note that the instrument used is, as stated by the authors, a system of categories. Again, this is a new term. Does this mean that categories are key performance indicators? Are they game situations? Are they patterns? Are they shots? The confusion continues and it does not seem to be clarified.

Regarding the validation of the instrument, the reviewer is unsure if the feedback of two tennis experts is enough to validate such a tool.

Furthermore, in the table 1 the confusion continues because the categories are then called “criteria”… thus not helping to clarify the overall confusion of terminologies.

Results

Again, when the authors state that they are analyzing technical-tactical performance indicators, there are several major issues: as indicated, the service is not a performance indicator, in the same way that the zone of the court in which the ball bounces, or where the ball is played to…

The titles, such as the probability of winning a point based on different combinations of performance indicators is again misleading, because the serve is not a performance indicator, nor the court surface, or zone of ball bounce…

Discussion

As indicated above, the results of these previous studies should be used as the main base to generate not only the structure of the results but also that of the discussion. This would help to provide the necessary clarity to the study and to show that the authors have done the adequate homework in their literature study by extracting the main results of these previous works and, once they have their own results, discussing adequately with those of their colleagues.

Conclusion

Again, the authors refer to the service, surface, and rally length. Two of these aspects are not key performance indicators, and this should be clarified.

The practical application section is extremely short, quite poor in content and suggestions for practitioners and with an obvious lack of belief in the applicability of the results obtained. It would be convenient to cover this with more depth and with a clear intention of sharing the positive aspects of the study.

Final words

Again, as in the first review, the reviewer would like to thank the authors for their interest in conducting studies on tennis. The reviewer hopes that the comments made in this review on the second version of the manuscript will be useful to the authors.

The fact that so many changes have been made has, on the one hand, helped to improve the English. However, on the other hand, new content, expressions, and terms have been introduced which may have contributed to confusion and opacity of the text.

The reviewer hopes that the authors receive these comments on the second version of the text as an attempt to improve the quality of the text and as part of the process involved in any scientific publication.

Once again, the reviewer's intention is to help the manuscript improve its quality and reach the appropriate level to be published in the journal. At the moment, this reviewer's recommendation is that the authors make considerable improvements to it.

Best regards,

Miguel Crespo, PhD.

International tennis Federation

Reviewer #2: The authors did a generally good job addressing the points of the initial review and the data on elite male players in singles tennis are valuable. They did not cite all the recommended articles or the older COMPUTENNIS data/Talbert & Old texts on patterns of tennis play. There is a need for clarity in focus on elite males singles (I presume singles rather than both singles and doubles) and reporting of “percentage points” or actual percentage differences. The revision still does not clearly communicate that the patterns and error rates are specific only to elite, male tennis players in singles match play. Please consider the revisions noted in the specific comments that follow.

Page-Line

2-1 . . . in elite men’s singles tennis

2-9

2-20 . . . on match strategy at the highest levels of elite men’s single tennis.

4-63 There should not be any one-sentence paragraphs. Consider revising to have a sentence with both objectives of tactical performance indicators and probability of winning at point in elite-level men’s tennis followed by a statement of importance. Something like: Data from the late rounds of three of four Grand Slam men’s singles tournaments allow the documentation of the tactical demands and success rates at the highest level of play of the sport.

14-245 to 311 Please revise carefully to clarify when you are talking about differences in error rates/percentages. For example, the wording should be: . . . “the probability of winning a point decreases 25 to 29 percentage points” if you are just subtracting error rates. When you calculate a percentage it is essential to know what the denominator is for the percentage calculation. So if you do not define this and just are subtracting percentages in different conditions, you have to clearly state that there is X percentage point difference, NOT at some percentage difference using some unspecified comparison condition

14-245 to 248 Delete redundant paragraph repeating the purpose of your study. Lead your discussion off with the most important observation(s) and their size/strategic meaning. Again, the author(s) should note that the results represent the highest level of performance in the sport—near and earning championship at the men’s elite leve1

14-249 to 253 Good comparison, but can there be more context added? Are previous studies of advanced or elite levels and both sexes?

17-315 . . . to the first serve in elite men’s singles tennis. Other option, perhaps better option, is to set up these conclusions from your data that relate to elite men’s singles tennis play.

17-318 . . . most common point winning combination of shot for the server was . . .

17-324 Revise to eliminate “guidelines.” You have general benchmarks for elite-level men’s tennis. You have no idea how different skill levels, disabilities, or sex of player might influence the pattens of play and error rates

17-329 How should coaches use this? Just say that these patterns and success/errors rates are what are currently present at the highest levels. Male players seeking this level of play should consider training and shots in match play that align with greater probability of success

18-344 Add a sentence that the stroke patterns and error rates observed may not be similar for players from other sex, skill level, or disabilities

7. PLOS authors have the option to publish the peer review history of their article (what does this mean?). If published, this will include your full peer review and any attached files.

Reviewer #1: **Yes: **Miguel Crespo

Reviewer #2: No

---

## [Decision Letter · Decision Letter 2]

9 May 2023

Match analysis and probability of winning a point in elite men's singles tennis

PONE-D-22-33800R2

Dear Dr. Prieto-Lage,

We’re pleased to inform you that your manuscript has been judged scientifically suitable for publication and will be formally accepted for publication once it meets all outstanding technical requirements.

Kind regards,

Javier Abián-Vicén, Ph.D.

Academic Editor

PLOS ONE

Reviewers' comments:

Reviewer's Responses to Questions

**Comments to the Author**

1. If the authors have adequately addressed your comments raised in a previous round of review and you feel that this manuscript is now acceptable for publication, you may indicate that here to bypass the “Comments to the Author” section, enter your conflict of interest statement in the “Confidential to Editor” section, and submit your "Accept" recommendation.

Reviewer #1: All comments have been addressed

Reviewer #2: All comments have been addressed

2. Is the manuscript technically sound, and do the data support the conclusions?

Reviewer #1: Yes

Reviewer #2: Yes

3. Has the statistical analysis been performed appropriately and rigorously? 

Reviewer #1: Yes

Reviewer #2: Yes

4. Have the authors made all data underlying the findings in their manuscript fully available?

Reviewer #1: Yes

Reviewer #2: Yes

5. Is the manuscript presented in an intelligible fashion and written in standard English?

Reviewer #1: Yes

Reviewer #2: Yes

6. Review Comments to the Author

Reviewer #1: The reviewer would like to thank the authors for considering most of the comments made. They have adequately addressed all the suggestions made. Finally, the reviewer would like to congratulate the authors for the considerable improvement of the manuscript.

Reviewer #2: Most all critiques have been addressed but one. Twice the authors have ignored the prompt to clearly point out that the relatively recent trust to automate, report, and analyze tennis tactical data is NOT the first attempts at gathering this knowledge. I know that PLOS One does not emphasize the need to provide clear justification/contribution for studies, but published reports should not OMIT or be dishonest about the novelty of ideas and scholarship. I find it frustrating that the editors keep requesting revisions but the authors to not, AT LEAST, note that the current data are just more recent efforts that follow initial work in the 1960s-70s (Talbert & Old) that were expanded by use of personal computers (COMPUTENNIS). A quick search for the latter turns of over a half dozen studies of tennis performance results using COMPUTENNIS.

7. PLOS authors have the option to publish the peer review history of their article (what does this mean?). If published, this will include your full peer review and any attached files.

Reviewer #1: **Yes: **Miguel Crespo

Reviewer #2: No

---

## [Editor Report · Acceptance letter]

16 May 2023

PONE-D-22-33800R2 

Match analysis and probability of winning a point in elite men's singles tennis 

Dear Dr. Prieto-Lage:

I'm pleased to inform you that your manuscript has been deemed suitable for publication in PLOS ONE. Congratulations! Your manuscript is now with our production department. 

Kind regards, 

on behalf of

Dr. Javier Abián-Vicén 

Academic Editor

PLOS ONE